# First Measurement of Soil Freeze/Thaw Cycles in the Tibetan Plateau Using CYGNSS GNSS-R Data

**Xuerui Wu** [1,2,3,*,†] , **Zhounan Dong** [2,3,†] , **Shuanggen Jin** [2,3] , **Yang He** [4] , **Yezhi Song** [2,3] , **Wenxiao Ma** [2,3] **and Lei Yang** [5]

1   School of Resources, Environment and Architectural Engineering, Chifeng University, Chifeng 024000, China
2   Shanghai Astronomical Observatory, Chinese Academy of Sciences, Shanghai 200030, China;
    zndong@shao.ac.cn (Z.D.); sgjin@shao.ac.cn (S.J.); syz@shao.ac.cn (Y.S.);
    mawenxiao19@mails.ucas.edu.cn (W.M.)
3   University of Chinese Academy of Sciences, Beijing 100049, China
4   School of Mechatronic Engineering and Automation, Shanghai University, Shanghai 200444, China;
    heyang2014@whu.edu.cn
5   College of Information Science and Engineering, Shandong Agricultural University, Taian 271018, China;
    yanglei@sdau.edu.cn
*   Correspondence: xrwu@shao.ac.cn; Tel.: +86-21-34775291; Fax: +86-21-64384618
†   The authors are equally contributed.

**Abstract:** The process of soil freezing and thawing refers to the alternating phase change of liquid water and solid water in the soil, accompanied by a large amount of latent heat exchange. It plays a vital role in the land water process and is an important indicator of climate change. The Tibetan Plateau in China is known as the "roof of the world", and it is one of the most prominent physical characteristics is the freezing and thawing process of the soil. For the first time, this paper utilizes the spaceborne GNSS-R mission, i.e., CYGNSS (Cyclone Global Navigation Satellite System), to study the feasibility of monitoring the soil freeze-thaw (FT) cycles on the Tibetan Plateau. In the theoretical analysis part, model simulations show that there are abrupt changes in soil permittivities and surface reflectivities as the soil FT occurs. The CYGNSS reflectivities from January 2018 to January 2020 are compared with the SMAP FT state. The relationship between CYGNSS reflectivity and SMAP soil moisture within this time series is analyzed and compared. The results show that the effect of soil moisture on reflectivity is very small and can be ignored. The periodic oscillation change of CYGNSS reflectivity is almost the same as the changes in SMAP FT data. Freeze-thaw conversion is the main factor affecting CYGNSS reflectivity. The periodical change of CYGNSS reflectivity in the 2 years indicates that it is mainly caused by soil FT cycles. It is feasible to use CYGNSS to monitor the soil FT cycles in the Tibetan Plateau. This research expands the current application field of CYGNSS and opens a new chapter in the study of cryosphere using spaceborne GNSS-R with high spatial-temporal resolution.

**Keywords:** GNSS-R; SMAP; CYGNSS; soil freeze and thaw cycles; soil moisture

---

## 1. Introduction and Background

The process of soil freezing and thawing refers to the alternating phase change of liquid water and solid water in the soil, which plays a vital role in the land surface hydrological process. The freeze-thaw cycle accompanies a large amount of latent heat exchange impacting the thermal energy of the soil, which in turn acts on the surface energy and water balance, so it is also an extremely important indicator of climate change. The area of the land surface undergoes annual freeze-thaw changes that can reach 50 million-$km^2$, mainly occurring in the northern hemisphere areas with latitude

over 40° and altitude above 1000 m [1]. The Tibetan Plateau is known as the "roof of the world" with an average altitude of more than 4000 m. Since it is the highest plateau in the world, it is called the "third pole" on Earth [2]. The Tibetan Plateau is also known as the fragile and sensitive area to global climate change. Therefore, accurately monitoring the surface freeze and thaw (F/T) state of the soil on the Tibetan Plateau not only benefits the local environment and climate studies but also contributes to the global climate change research [3,4].

Traditional methods to monitor the surface F/T state can be divided into field monitoring and numerical simulation. Although the former can provide more accurate and reliable results in the study area, the natural environment of the Tibetan Plateau is too poor to deploy many in-situ monitoring stations. The established field stations are relatively sparse and most of them are concentrated in a small area. Therefore, it eagerly needs to improve the spatial resolution of the F/T state dataset in this region to contribute to the related hydrology and climate research [5]. The numerical simulation method generally can cover the large-scale area, and it generally includes a large number of parameters to be determined, introducing or ignoring a certain parameter can greatly impact the simulated results, which means the results do have a certain degree of uncertainty. Besides, the simulation model also relies on the measured data to some extent and has a low spatial resolution. The development of satellite remote sensing technology provides a new approach for surface F/T monitoring. Since the visible light and thermal infrared remote sensing are easily blocked by clouds, while microwave remote sensing can work at all-day and all-weather conditions, such as the polar-orbit satellite of European Space Agency's (ESA) Soil Moisture and Ocean Salinity (SMOS) mission and National Aeronautics and Space Administration's (NASA) Soil Moisture Active Passive (SMAP) mission. It is the most effective way for surface F/T monitoring, especially in high altitude cold or remote unmanned areas [6–8]. However, the drawback of traditional monostatic active and passive satellite-based scatterometer or radiometer is that it is difficult to combine higher spatial and temporal resolution, which cannot accurately determine the F/T change time within a small area.

Global Navigation Satellite System-reflectometry (GNSS-R) is an innovative Earth observation technology using the signal of opportunity (SoOP) for terrestrial remote sensing; it has been successfully applied in many geoscientific fields in recent years [9–11]. GNSS-R directly employs the Earth's surface bouncing off signal transmitted from GNSS satellites to remotely sense geophysical parameters of interface for Earth observation [12,13], such as sea surface wind speed detection, ocean altimetry, soil moisture, forest biomass, inland water, and wetlands [14–18]. Compared with the monostatic remote sensing satellite, the GNSS-R platform just need to carry delay/Doppler receiver with low mass and power consumption; this greatly reduces the satellite-based GNSS-R deployment cost. Therefore, it allows the project affords a specific constellation providing continuous observation at the global scale with the high spatial and temporal resolution in a low-cost way. Cyclone Global Navigation Satellite System (CYGNSS) mission sponsored by NASA is the first specific GNSS-R constellation and is launched in December 2016 [19]. CYGNSS aims to monitor the wind speed near the central area of a tropical cyclone and to help the study on forecasting the intensity of typhoons. GNSS-R receiver simultaneously receives direct signals for positioning and incoming signal power calibration. In addition to CYGNSS, several spaceborne GNSS-R missions have been successfully deployed; the first GNSS-R test satellite is United Kingdom-Disaster Monitoring Constellation (UK-DMC) launched in 2004. By 2014, the UK Technology Demonstration Satellite-1 (TDS-1) was successfully launched into orbit with SSTL's (Surrey Satellite Technology Ltd.) latest spaceborne GNSS-R hardware receiver to test the new generation of Delay/Doppler Mapping Instrument for CYGNSS mission. In June 2019, China launched its first dedicated GNSS-R mission Bufeng-1 A/B [20]. The spaceborne GNSS-R payload Global Navigation Satellite System Occultation Sounder II (GNOS II) developed by the National Space Science Center of the Chinese Academy of Sciences is expected to be launched in later 2020 [21]. Although the main scientific objective of these missions is to measure the wind field for ocean applications, it also provides a great opportunity for land surface parameters and the cryosphere study.

Although lots of work related to soil F/T detection using the traditional microwave remote sensing has been done [22–24], using the GNSS-R technology for surface F/T state monitoring is still a new application field [25,26]. The previous study based on the theoretical analysis has verified that using the GNSS-R/IR technology for surface freezing and thawing state monitoring is feasible. The tracking dataset of Plate Boundary Observatory (PBO) stations is used for ground-based preliminary verification, and the correlation between the surface freeze-thaw state and multipath observation has been established [27]. Comite et al. used the TDS-1 datasets to conduct a preliminary analysis of surface freeze-thaw state monitoring at high latitudes; since the revisit time of the TDS-1 satellite is longer, only the monthly results are presented [28]. Among the existing spaceborne GNSS-R missions, only the CYGNSS can provide successive and high spatial-temporal resolution observation for land remote sensing. In this study, time series from January 2018 to January 2020 of CYGNSS datasets are employed to validate the potential of GNSS-R for surface freeze-thaw monitoring.

The rest of the paper is organized as follows. Section 2 mainly introduces the theoretical fundamentals. Section 3 shows the dataset and processing methodology. Discussion and conclusions are presented in Sections 4 and 5, respectively.

## 2. Theoretical Fundamentals

The theoretical fundamental of surface F/T state monitoring is that the complex permittivity of liquid water and other natural substances are different. In the microwave band, it is very sensitive to the change of liquid water in the near-surface soil (5 cm depth to the topsoil). As the temperature decreases, most of the liquid water in the soil undergoes a phase change and it converts into solid ice, thus leading to an abrupt decrease in the complex permittivity of the soil. At the same time, the complex permittivity is related to the reflectivity.

### 2.1. Mixed Permittivity Model

To verify the response of complex permittivity to the surface soil F/T state, the permittivity model of the mixed medium is used at different soil temperatures in this study. Above freezing temperature, soil can be regarded as a dielectric mixture composed of air, solid soil particles, free water, and bound water. The permittivity of the mixed medium of soil and water $\varepsilon_s$ can be formulated as follows [29,30]:

$$\varepsilon_s \ = \ 1 + \frac{\rho_b}{\rho_s}(\varepsilon_s^\alpha - 1) + m_v \varepsilon_{fw}^\alpha - m_v \tag{1}$$

where $\rho_b$ is the bulk density. $\rho_s$ is the solid density and $\varepsilon_s^\alpha$ and $\varepsilon_{fw}^\alpha$ are the permittivity of the solid matter and pure water, respectively. $\alpha$ is the shape factor. For frozen soil, the phase state of the water in the soil changes, and the permittivity model adds the calculation of the ice composition. The final permittivity of frozen soil can be expressed as follows [31]:

$$\varepsilon = V_s \varepsilon_s^\alpha + V_a \varepsilon_a^\alpha + V_{fw} \varepsilon_{fw}^\alpha + V_{bw} \varepsilon_{bw}^\alpha + V_i \varepsilon_i^\alpha \tag{2}$$

where $V$ is the volume content of different components, the subscripts $s$, $a$, $fw$, $bw$, and $i$ are solid soil, air, free water, bound water, and ice, respectively. Using the above permittivity models, Figure 1 simulates the relationship between soil complex permittivity and soil temperature under different soil moisture contents. It can be seen that for different soil moisture contents when the surface soil temperature is greater than 0 °C, the real and imaginary parts of the permittivity have a significant sensitivity to the change of soil moisture. On the contrary, the complex permittivity of soil changes slightly below freezing temperature, especially for the imaginary part, it keeps the same values. This is mainly because the water in the soil changes from solid to liquid. The changing characteristics of the permittivity are the theoretical basis for surface freeze and thaw monitoring in the microwave band.

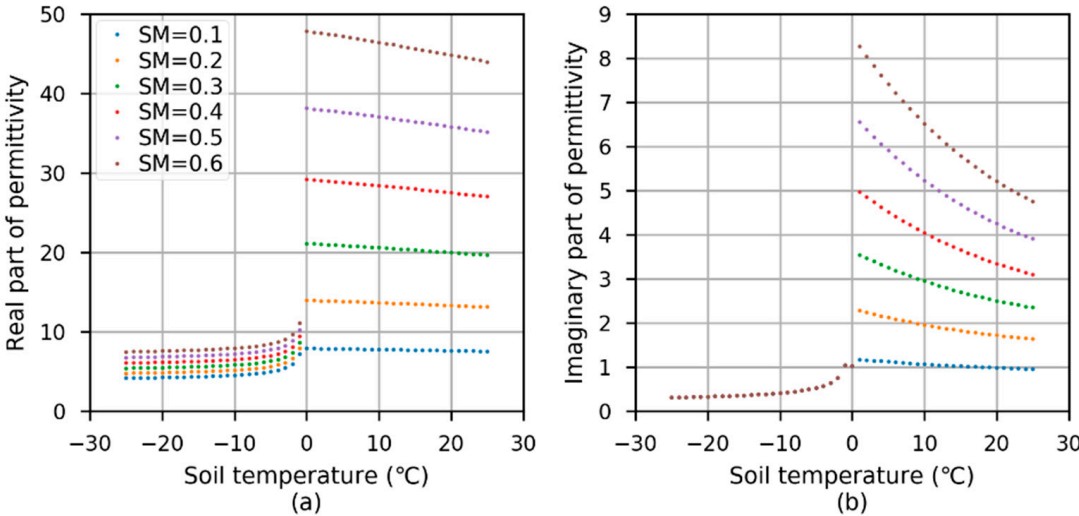

**Figure 1.** The relationship of the real part (**a**) and imaginary part (**b**) of permittivity and soil temperature under different soil moisture contents. SM stands for soil moisture, while its unit is cm$^3$/cm$^3$. Soil texture for the simulations is: sandy percentage is 51.5 %, clay percentage is 13.5 %, while the bulk density is 1.60 g/cm$^3$.

### 2.2. Surface Reflectivity at LR Polarization

To overcome the Faraday rotation effects, the navigation satellite transmits the right-hand circularly polarized (RHCP) signal. Since the surface reflection can change the polarization characteristics, most of the scattering signals are left-hand circular polarization (LHCP). GNSS-R antenna commonly uses LHCP which is different from the zenith positioning antenna to receive reflection signals. The surface reflectivity Γ is the function of the Fresnel reflection coefficient, radar geometry, and surface roughness [32,33].

$$\Gamma(s, \theta_i, \varepsilon) = \left| \Re_{LR}(\theta_i, \varepsilon) \right|^2 \cdot exp\left(-4\psi(s, \theta_i)^2\right) \tag{3}$$

where $\Re_{LR}$ is the Fresnel reflection coefficient at LR polarization, and it is the function of incidence angle and permittivity. $\theta_i$ is the signal incidence angle. $\psi$ is the roughness factor, which is the function of the surface root-mean-square (RMS) height $s$ and wavenumber $k$ ($\psi = kscos\theta_i$). The Fresnel reflection coefficient at LR polarization can be calculated as follows [32,33]:

$$\Re_{LR}(\theta_i, \varepsilon) = \frac{cos\theta_i(\varepsilon - 1)\sqrt{\varepsilon - sin^2\theta_i}}{\left(\varepsilon cos\theta_i + \sqrt{\varepsilon - sin^2\theta_i}\right)\left(cos\theta_i + \sqrt{\varepsilon - sin^2\theta_i}\right)} \tag{4}$$

where $\varepsilon$ can be calculated by the aforementioned mixed medium permittivity models. Figure 2 simulates the relationship between reflectivity and incidence angle under different temperatures. The soil texture information is the same as presented in Figure 1. The permittivities used to calculate the surface reflectivities are received from the models shown in Section 2.1. The soil moisture values used in Figure 2a–c are 0.1, 0.2, and 0.3 cm$^3$/cm$^3$, respectively. Over the wet area, the soil temperature positive–negative conversion can greatly change the land surface permittivity which can cause 2 dB change in reflectivity when the soil moisture is 0.3 cm$^3$/cm$^3$; this effect weakens as the soil moisture decrease. The incidence angle can also affect the reflectivity, especially at the high incidence angle.

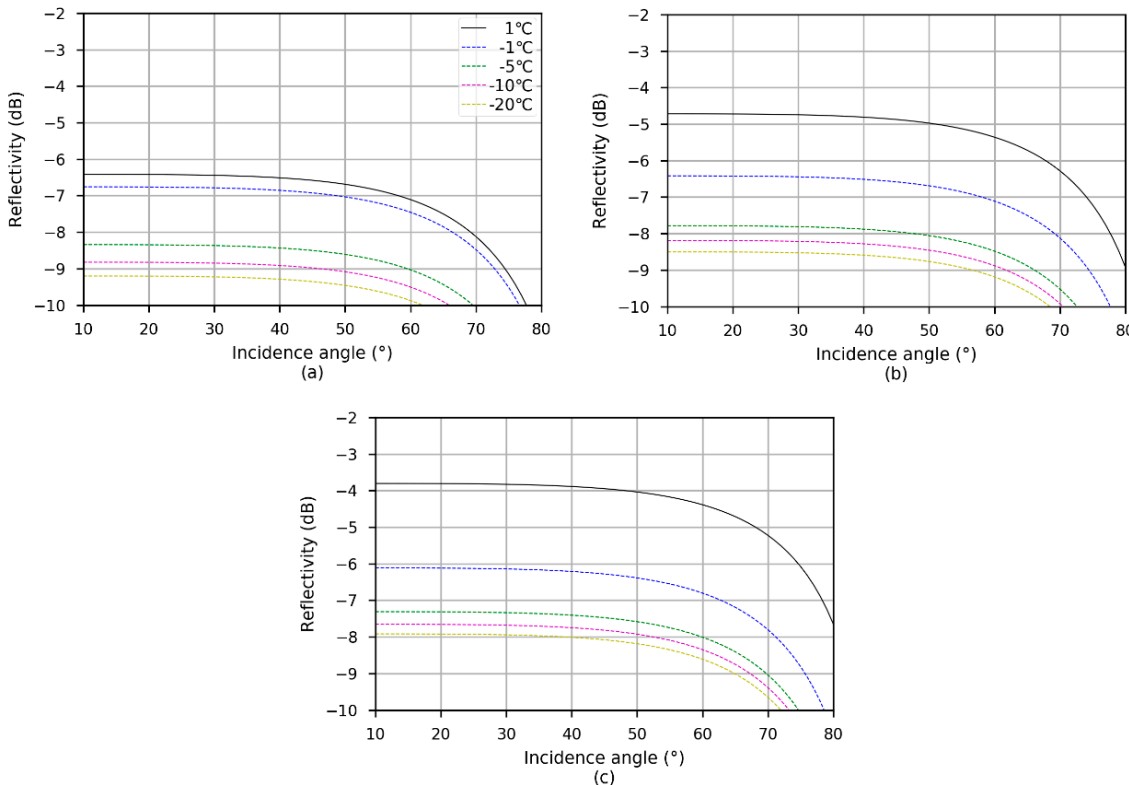

**Figure 2.** The relationship of soil reflectivity and incident angle under different temperature with soil moisture 0.1 cm$^3$/cm$^3$ (**a**), 0.2 cm$^3$/cm$^3$ (**b**), and 0.3 cm$^3$/cm$^3$ (**c**).

Assuming the coherent component dominates GNSS-R land echoes, the incoherent contribution is negligible. The former only comes from the first Fresnel zone; the scattering power can be obtained through the Friis transmission equation [33].

$$P_{coh} \;=\; \frac{P_T G_T \lambda^2 G_R}{(4\pi)^2 (R_R + R_T)^2} \Gamma(s, \theta_i, \varepsilon) \tag{5}$$

where $P_T$ is the GNSS satellite transmit power. $G_T$ is the GNSS satellite antenna gain at specular point direction. $P_T G_T$ is the GNSS Equivalent Isotropically Radiated Power (EIRP). $\lambda$ is the wavelength. $G_R$ is the receiver antenna gain. The distance between the receiver/transmitter and the specular point is presented as $R_R$ and $R_T$, respectively. According to Equation (5), the reflectivity can be formulated as follows [33]:

$$\Gamma(s, \theta_i, \varepsilon) \;=\; \frac{(4\pi)^2 P_{coh}(R_R + R_T)^2}{\lambda^2 G_R P_T G_T} \tag{6}$$

Compared to the TDS-1 mission published datasets, the CYGNSS team has done a lot of work for calibration and unwrapping the observations and also build the GPS transmit monitoring system; the necessary auxiliary data in Equation (6) have been included in the published dataset.

## 3. Dataset and Processing Methodology

In this study, four public datasets are used to validate the potential of CYGNSS to monitor the soil freeze and thaw cycle on the Tibetan Plateau: 1) CYGNSS Level-1 data products, 2) SMAP Level-3 FT fraction (soil moisture active and passive), 3) IGBP (International Geosphere-Biosphere Programme) land cover classification from MODIS data with the spatial resolution of 1 km, and 4) ERA5 surface soil moisture and soil temperature. Here, we will give more details about the process of CYGNSS datasets.

CYGNSS was successfully launched in December 2016. The revisiting time is 2.8 (median) and 7.2 (mean) hours per day. The theoretical footprint of a reflected GNSS signal is about $0.5 \times 7.0$ km. CYGNSS consists of 8 small satellites equipped with GNSS-R receivers. Each small satellite can simultaneously record the reflection signal of the L1 carrier frequency of 4 GPS satellites. The CYGNSS receiver has four channels and can generate 4 DDMs (Delay Doppler Map) at a time. DDM is a function of scattering geometry, antenna gain, distance, surface dielectric, and statistical characteristics, and it is the CYGNSS observable.

In general, the reflected signals of GNSS constellation are composed of coherent and noncoherent components. At present, the calculation of coherent and noncoherent components of CYGNSS is an open issue. For CYGNSS soil moisture inversion, in the range of GNSS-R footprint, the coherent energy mainly comes from the specular reflection of inland water, which will cause the peak value of DDM to increase. The peak energy of the coherent part will be many times larger than the incoherent component. Therefore, as for the soil moisture inversion, only considering the coherent reflection is very disadvantageous. The study found that there is a strong correlation between the WAF (Woodward ambiguity function) of the coherent reflection and the WAF of the noncoherent reflection. DDM greater than the threshold is considered to be mainly from coherent scattering and is eliminated [33].

However, most studies believe that the receiver mainly receives the coherent scattering power from the image of the first Fresnel zone [34,35]. For the coherent part, the DDM can be expressed based on the Friis transmission formula and the Fresnel reflection coefficient of the equivalent smooth surface, as shown in the above equation. As for the soil F/T cycles analysis in our paper, we adopt that the energy of the CYGNSS data is coherent, there is no consideration of noncoherent scattering in our analysis.

In the case of the same soil moisture and vegetation cover, the angle is a major factor affecting the reflected power of the GNSS constellation. The processing of the angles is also an open issue in the CYGNSS study. The previous observation satellite systems have a single angle of incidence or a small range of incidence angles. Differently, CYGNSS can receive a wide range of incident information (incidence angle from 0° to 70° with a standard deviation of 16.70°). In the literature [35], the observation data of each angle at the pixel is normalized by the data within the angle range of 35° ± 5° within the same pixel. Considering the data quality, the incidence angles greater than 60° are removed, and the data in the nadir direction is used for normalization processing [33]. In the data processing of this article, we adopt the methodology presented in the literature [33].

Figure 3 shows the CYGNSS specular point reflectivity in the Qinghai-Tibet Plateau on January 1, 2018. It can be seen that for the very north part of Qinghai-Tibet Plateau, there are sparse specular points, but in the middle part and the south part, the distribution of the specular points is denser. Blue parts are water bodies, which should be masked during our analysis to remove possibly contaminated reflections.

To eliminate the influence caused by the surface heterogeneity, the IGBP land classification data is used to distinguish the research area into four types (Figure 4), i.e., mixed-forest, open shrubland (desert), grassland, and barren/desert. In the subsequent analysis, it will be carried out in the relatively homogeneous land cover class. In this way, we can ensure that the features in each zone are relatively uniform so that the changes in the CYGNSS observations caused by different types of surface coverage can be excluded. Now, our analysis can focus on surface FT changes.

*Comparison between CYGNSS Daily Reflectivity with the SMAP FT State*

Here, we demonstrate the feasibility of using space-borne CYGNSS data for surface freeze-thaw monitoring. In the Tibetan Plateau, the CYGNSS daily reflectivity is compared with the SMAP FT fraction and FT state data. The time-series span is from January 2018 to January 2020. During this time range, the daily CYGNSS reflectivity is compared with the soil F/T state received from SMAP L3 data.

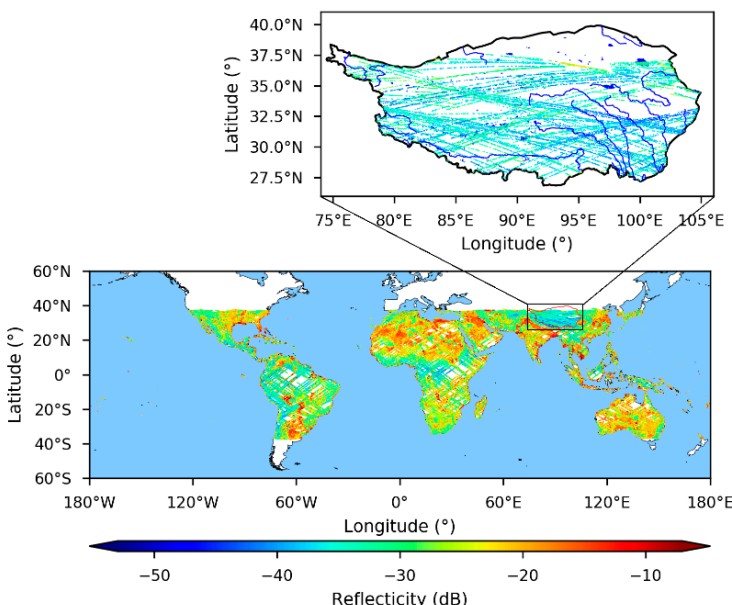

**Figure 3.** Cyclone Global Navigation Satellite System (CYGNSS) specular point reflectivity in the Qinghai-Tibet Plateau on January 1, 2018. Blue areas are water bodies.

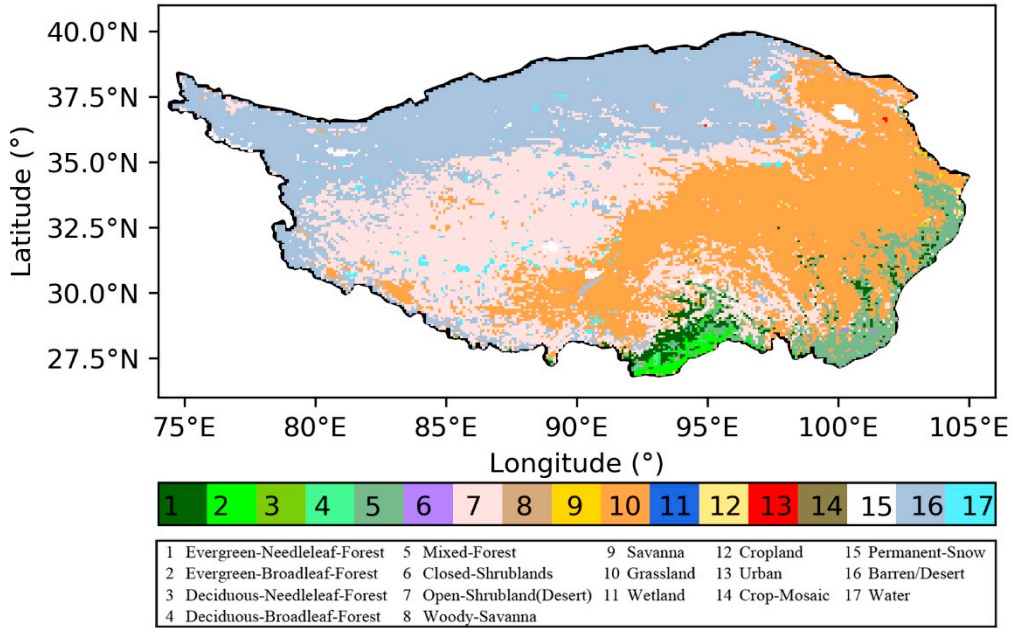

**Figure 4.** The International Geosphere-Biosphere Programme (IGBP) land cover type on the Tibetan Plateau.

In each classified area, we use the CYGNSS L1 data to calculate the daily reflectivity, and the SMAP L3 data are used to calculate the percentage of frozen pixels. Figure 5 presents the time series of surface reflectivity versus SMAP freeze ratio area under different land cover types on Tibetan Plateau. From the simulation, we can see that the oscillation of the frozen pixel ratio is, in general, consistent with the daily CYGNSS reflectivity. Soil frozen time mainly occurs from October to April in the second year. For this time range, water in the soil will become solid ice to a certain degree, which will make the surface reflectivity decrease accordingly. When the soil begins to thaw, the water in the soil become liquid, which will increase the permittivity at L band and thus, leads to the increase of surface reflectivity. On the whole, the trends plotted the time-series changes in Figure 5 meets the above discipline. The remaining problem for the abnormal oscillation is thought to be caused by the two factors, one is the vegetation volume scattering, which will attenuate the coherent scattering,

and another one is perhaps due to the data process uncertainties. However, in general, the analysis results are enough to prove that the CYGNSS daily reflectivity can be used for freeze-thaw cycle monitoring in Tibetan Plateau.

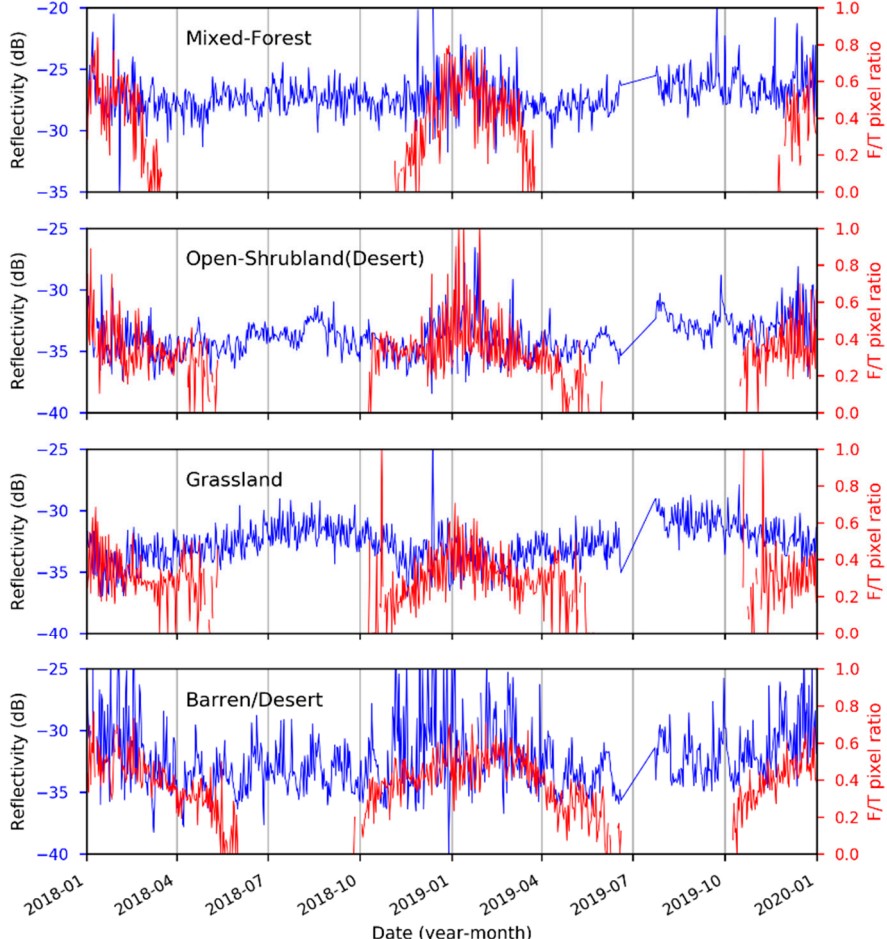

**Figure 5.** The time series of surface reflectivity versus Soil Moisture Active Passive (SMAP) provided freeze ratio area under different land cover types on Tibetan Plateau.

It is well known to us that soil moisture will affect the surface reflectivity. To take this effect into considerations, we also plot the time series of average surface reflectivity versus SMAP average soil moisture under different land cover types on Tibetan Plateau. From Figure 6, we can see that the dynamic range of moisture content is very small, lower than 0.3. Except for the grassland, the volumetric soil moisture content is between 0.1 and 0.2. Thus, a small dynamic range of soil moisture will have little effect on the CYGNSS reflectivity. We can deduce that the oscillations of the CYGNSS daily reflectivity in Figure 6 are mostly caused by the soil F/T change.

To give a more self-evident illustration, Figure 7 presents the monthly average surface soil moisture (January and July in Figure 7a,c) and the corresponding soil moisture contents in 2018. From Figure 7a,b, we can see that for these two months, the changes in soil moisture are now obvious. Compared with Figure 7a,b, Figure 7c,d show very obvious soil temperature changes for these two months. However, the soil temperature as illustrated in Figure 7c,d has varied from below 0 °C to above 0 °C for almost the whole area of the Tibetan Plateau. This further proves that the change of the surface reflectivities in these two months is caused by the freezing/thawing conversion of the soil.

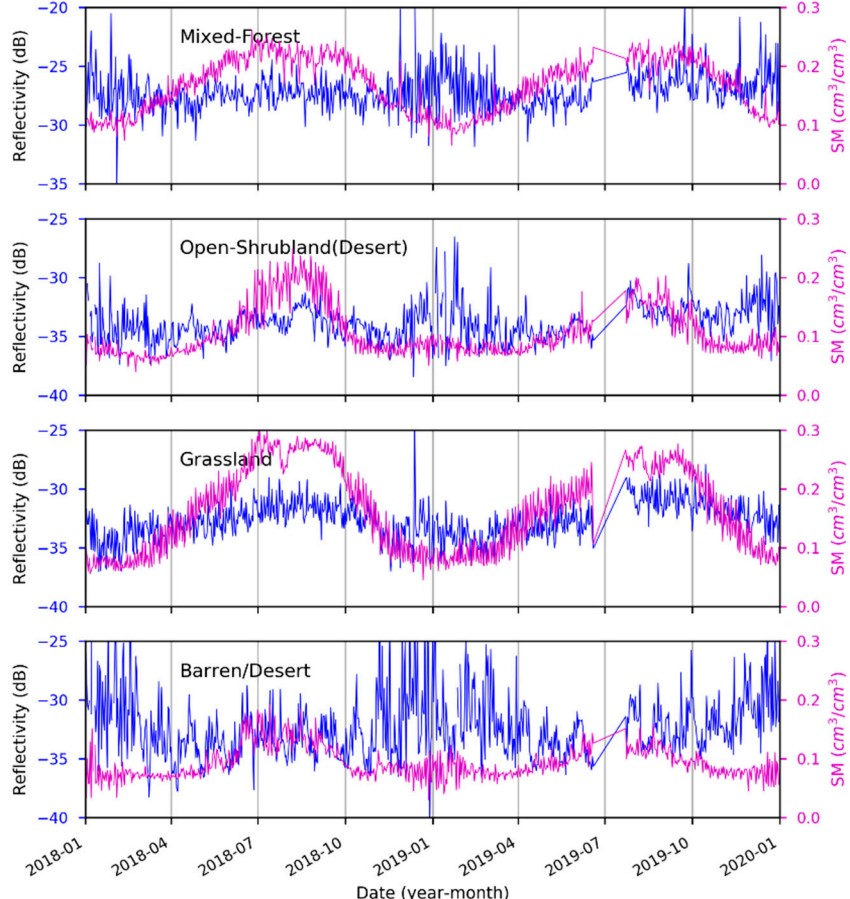

**Figure 6.** The time series of average surface reflectivity versus SMAP average soil moisture under different land cover types on Tibetan Plateau.

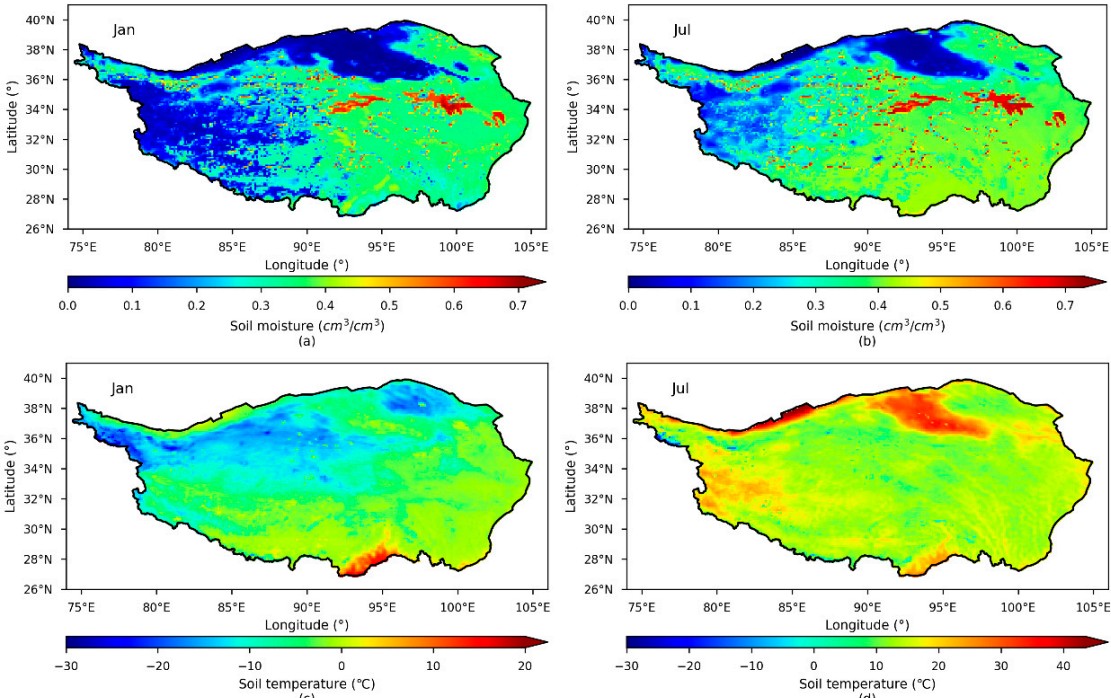

**Figure 7.** The January and July monthly average surface soil moisture (**a**,**b**) and soil temperature (**c**,**d**), respectively, on Tibetan Plateau in 2018.

## 4. Discussion

GNSS-R has emerged as a promising remote sensing technique. However, it is still a very new application for soil FT detection. This article explores the feasibility of using CYGNSS for surface FT monitoring for the first time. CYGNSS data are publicly available to users since March 2017. Due to data quality problems in the Tibetan Plateau, we selected data from January 2018 to January 2020 for analysis. However, it is unable to obtain the field measurement data for this time series in Tibetan Plateau, so we chose SMAP FT data for comparative analysis. Overall, our analysis has shown that it is feasible to use the CYGNSS data for soil FT detection. This opens a new door for the cryosphere to analyze the soil surface freeze-thaw state with high spatial and temporal resolution.

However, there are many open issues in CYGNSS data processing. One is the processing of surface coherent items and noncoherent items and the other is the processing of scattering geometry. In this paper, the current methods commonly used in CYGNSS soil moisture retrievals are adopted: it is believed that the received energy mainly comes from surface coherent scattering, and the angle processing method adopts the normalization processing method given in the literature [30]. In the follow-up analysis, we will enhance the ability of data processing methods and strive to reduce the error determination of soil FT state caused by data processing.

This paper qualitatively shows that the use of CYGNSS for soil FT detection is feasible. However, no specific discriminant algorithm is given, and important parameters of soil FT studies such as freezing time, thawing time, and freezing period are not given. These are all problems that need to be refined in the subsequent research.

At present, the dual microwave algorithm, decision tree algorithm, and discriminant function algorithm have been developed for the discriminant algorithm of passive microwave on the surface freeze-thaw state. These algorithms are based on the use of frequency and polarization information to expand the judgment. Based on this idea, with the acquisition of multifrequency data from spaceborne GNSS-R and the design of multi-polarization (circular polarization and linear polarization) antennas, these will provide more effective information on the surface, so the development of multifrequency and multi-polarization surface FT discriminant algorithms for GNSS-R systems will be an important direction for the subsequent discriminant algorithm development.

Besides, during our analysis, the Tibetan Plateau is classified into four different research areas to overcome the impact of different surface types. At the same time, the influence of soil moisture on freeze-thaw characteristics was considered. In future research, snow cover is an important factor that must be considered. The direct and reflected signals of navigation satellites will cause volume scattering, attenuation, absorption, etc. within the snow layer, and these will result in changes in the receiver signals. How to remove the influence of these factors is the focus of future research.

In the Tibetan Plateau region, vegetation changes little during the soil FT process. Therefore, we do not consider the effects of vegetation in our analysis.

The current latitude coverage of CYGNSS is between 38° north and south latitudes. The lack of coverage in high latitudes limits the application for soil FT detection. Spaceborne GNSS-R coverage in high latitudes will be the subsequent development direction. The coverage of high latitudes will be more conducive to the monitoring of surface FT state, making GNSS-R an effective and beneficial supplement to the existing polar-orbiting satellites, with higher spatial and temporal resolutions.

## 5. Conclusions

This paper investigates the potentiality of the soil FT detection using space-borne CYGNSS in Tibetan Plateau. In our analysis, the CYGNSS reflectivities spanning from January 2018 to January 2020 were selected to compare with SMAP FT data. The oscillation dynamic of CYGNSS reflectivities is very similar to SMAP FT data. In order to exclude the influence of soil moisture, this paper compares the soil moisture data with the CYGNSS reflectivities and finds that the influences of soil moisture are very small and can be ignored. Since the study areas are within homogeneous land cover classes, the oscillation of the daily reflectivity during the time series is thought to be caused by the effects of

the soil FT state, it is believed that the periodic variation of CYGNSS reflectivity over the 2 years is mainly due to soil FT changes. Our study demonstrates the feasibility of using CYGNSS for surface freeze-thaw monitoring in Tibetan Plateau.

**Author Contributions:** X.W. and Z.D.: idea proposal, conceptualization, and original draft; S.J.: suggestions and review; Y.H. and Y.S.: help download related dataset; W.M. and L.Y. suggestions. All authors have read and agreed to the published version of the manuscript.

**Funding:** This research was funded by the National Natural Science Foundation of China (No. 41501384 and No. 31971781).

**Conflicts of Interest:** The authors declare no conflict of interest.

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
