# Peer review of "First Measurement of Soil Freeze/Thaw Cycles in the Tibetan Plateau Using CYGNSS GNSS-R Data"

_remotesensing, doi:10.3390/rs12152361_

Round 1

Reviewer 1 Report

An interesting paper. It is essentially a technical exercise in processing CYGNSS data to explore whether freeze-thaw information is available in the data. The results show that it is, although section 4 then points out that snow and vegetation will influence the results and more research is needed.

The standard of English is poor.

GNSS stands for Global Navigation Satellite System (see page 2 line 69). DDM (section 3) needs an explanation. In Figure 3 it is Reflectivity not Refleticity. 

Author Response

Dear reviewer, 

   Thank you very much for your suggestions.

   They are very helpful for our improvement.

   More details for the revision, please see the attachment.

Best regards

Xuerui

Reviewer 2 Report

General comments:

This manuscript gave an example to identify the freeze-thaw cycles that were difficult and arduous to observe in fields by CYGNSS GNSS-R data. I'd like to encourage such attempts and recommend it to be accepted after revision.

Specific comments:

Abstract: The estimation accuracy and uncertainties should be quantitatively added in the key results.

Introduction: Some references should be added to justify the technical feasibility of using CYGNSS GNSS-R data to check soil freeze and thaw processes.

L 44-45: I can’t understand the logic between “which means”. “Fragile and sensitive” meant “driver”?

L 109-113: Repeated statement. It seems unnecessary.

Theoretical fundamentals: Any equation(s) cited here should add the reference(s).

L115-120: How to eliminate the effects of snowpack and/or ice on the soils?

L140: What is the data source of Figure 1? Cited from the previous studies or tested by the authors? Which kinds of soils? What are the testing conditions and scales? How to test the real and imaginary parts of the permittivity? SM meant “soil moisture”? No unit for SM?

L175: As my comments for Figure 1, more specific information should be added to explain its source, conditions etc.

Dataset and processing methodology:

L230: Which literature?

L241-242: What were the consequences of such a homogeneous treatment? Any inacceptable impact on the estimation results?

L247-251: As the authors said, for the drawback of SMAP not to accurately determine the F/T change time within a small area, how could the authors use such data to calculate the FT fraction and state data? Any field results to verify the estimation results?

L282-283: It’s discussion, not a result.

Discussion: Need further contents to discuss the reliability, sensitivity, uncertainty of the methodology. Both merits and shortcomings should be mentioned.

L292-297: Yes, the authors mentioned what I’m concerned. How to solve this problem? The methodology reliability of this manuscript might be weakened.

L299-302: I don’t think vegetation is a big problem, for they almost keep the similar status during the F/T periods in the Tibetan Plateau.

Conclusions:

Self-appraisal of the methodology proposed in this manuscript should be supported with objective results.

Author Response

Dear Reviewer,

   Thank you so much for your valueble suggetions, which are very helpful for our improvement. 

  According to your suggestions, we revised our manucript.

  Please see the attachment for more details.

Best regards,

Xuerui 

Round 2

Reviewer 2 Report

The authors have rephrased their discussion well; however, the other parts need further improvement. Not all my comments were responded, especially for the justification of verification.

L240-242: As the authors said, for the drawback of SMAP not to accurately determine the F/T change time within a small area, how could the authors use such data to calculate the FT fraction and state data? Any field results to verify the estimation results?

In addition, some mistakes should be corrected as well. For example, the unit of bulk density in the caption of Figure 1 .

Author Response

Dear Reviewer, 

   Thank you very much for your suggestions.

   We are sorry that we have not replied your comments clearly in Round 1.

   Now, we present our responses as given in the attachment.

   Best wishes,

Xuerui Wu
